# The Amount of Data Required to Recognize a Writer’s Style Is Consistent Across Different Languages of the World

**DOI:** 10.3390/e27101039

**Published:** 2025-10-04

**Authors:** Boris Ryabko, Nadezhda Savina, Yeshewas Getachew Lulu, Yunfei Han

**Affiliations:** 1Federal Research Center for Information and Computational Technologies, 6300090 Novosibirsk, Russia; 2Department of Information Technologies, Novosibirsk State University, 6300090 Novosibirsk, Russiaj.lulu@g.nsu.ru (Y.G.L.); yunfei.han@mail.ru (Y.H.)

**Keywords:** information technology, data compression, language family, language group, individual author’s style of the writer, information-theoretic method (RS-method), hypothesis testing

## Abstract

In this paper, we apply an information-theoretic method proposed by Ryabko and Savina (therefore called the RS-method), based on the use of data compression, to recognize the individual author’s style of a writer across four languages from different language groups and families. In this paper, the presented method was used to study fiction texts in Russian (East Slavic group of languages of the Indo-European language family), Amharic (South Ethiosemitic group of the Semitic language family), Chinese (Sinitic group of the Sino-Tibetan language family) and English (West Germanic language group of the Indo-European language family). It was found that the amount of data necessary for recognizing an author’s style is almost the same for all four languages, i.e., the amount of data is invariant across different language groups. The results obtained are of interest to computer science, literary studies, linguistics and, in particular, computational linguistics.

## 1. Introduction

### The Concept of the Individual Author’s Style of a Writer

An author’s style is a unique set of features characteristic of a particular writer’s work, which makes their novels recognizable and different from those of other writers [1,2,3]. An author’s style is formed during the creative process and reflects the individuality and worldview of the author [3,4,5,6]. In fiction, the styles of different writers are extremely diverse. For example, one can recall the businesslike and laconic style of Ernest Hemingway, the fussy James Joyce, the sardonically abrupt Kurt Vonnegut [2] or the heavy and cumbersome style of Tolstoy. An author’s style is formed gradually over the course of their life and reflects the evolution of the author [4].

In our work, we used the classification of sources of text variability proposed by the leading mathematician A.N. Kolmogorov, who is also known for his results in the field of information theory [7,8]. He identified the following sources of text variability: content, form and unconscious individual author’s style. Many researchers argue that individual author’s style is a reflection of the writer’s personality [1,2,3,4]. The elements of author’s style are well known. These include vocabulary (use of certain words and expressions), syntax (features of sentence construction), tropes and figures of speech (metaphors, epithets, comparisons, etc.) and composition (arrangement of parts of the work), as well as the general tone and mood of the work [3].

Studying the elements of author’s style requires multi-tasking and is a rather difficult problem. But recognizing the author’s style of a writer without pursuing analysis of style elements is a completely different task. The information-theoretic method proposed by Ryabko & Savina [9,10,11], which we call the RS-method, helps to solve this problem reliably, i.e., with the help of the apparatus of mathematical statistics. It is based on the use of so-called archivers or data compression methods, which, in turn, can be attributed to information theory. The fact is that modern archivers are aimed at finding a variety of patterns in compressed texts, including through using methods such as describing the text with the shortest formal grammar, building dictionaries of minimal volume describing the text and other methods related to artificial intelligence.

An important application of data compression for classification was proposed by P. Vitani and developed by him and his co-authors in several papers (see [12,13] and the references therein). They used the length of a compressed message as an estimate of its Kolmogorov complexity and, based on this, proposed the so-called normalized compression distance between two different texts. This approach made it possible to classify different human languages, animal species (based on their genomes), computer and biological viruses and some other objects. The main difference between the application of normalized compression distance and our approach is the integration of the latter with methods from mathematical statistics, which makes it possible to apply the developed apparatuses of this science, including hypothesis testing and numerical measures such as Cramer’s coefficient.

In this study, we applied this method of recognizing author’s style to four languages: English, Russian, Amharic and Chinese. This choice was due to the fact that these languages belong to different language groups and families. An unexpected result was obtained: author’s style was reliably determined in texts across such different languages using the same amount of data, measured in kilobytes and not in the number of letters, symbols or similar units. Thus, we can assert that the amount of data necessary to determine the author’s style of a writer is, in a sense, invariant for all the languages we have considered.

## 2. RS-Method for Recognizing the Author’s Style of a Writer

### 2.1. The Idea of the Method

The method of recognizing an author’s style is based on the use of algorithms for lossless compression, implemented in the form of so-called archivers. Their purpose is to encode texts in such a way that the length of the encoded message is shorter than the original (the text is compressed) and, if necessary, the encoded text can be decoded into the original. Text data was fed as the input of the archiver, which encoded the text data into files of shorter length, i.e., compression. Compression occurs because archivers find unevenness in the frequencies of occurrence of letters and words and use hidden patterns based on the theory of formal grammars and the laws of information transmission. Let us briefly describe the scheme of application of the developed method. Let us define three texts, T_1_, T_2_, T_3_, and it is known that T_1_ and T_2_ were generated by different sources of information, I_1_ and I_2_, and T_3_ was generated by either I_1_ or I_2_ (for example, T_1_ is a text in English, T_2_ is in German, and T_3_ is in English or German). Let d be some archiver, and, if it is applied to some file X, then the length of the compressed file is denoted by d(X). First, the texts are combined into the pairs T_1_T_3_ and T_2_T_3_, and both pairs are compressed. Then, we separately compress files T_1_ and T_2_, after which we calculate the differences in the lengths of the compressed files: d(T_1_T_3_) − d(T_1_) and similarly d(T_2_T_3_) − d(T_2_). If d(T_1_T_3_) − d (T_1_) is less than d(T_2_T_3_) − d(T_2_), then we conclude that the text T_3_ was generated by the information source I_1_. If d(T_1_T_3_) − d(T_1_) > d(T_2_T_3_) − d(T_2_), then T_3_ was generated by the information source I_2_. This conclusion is due to the fact that the archiver, when compressing later texts, i.e., T_3_, uses the statistical features it found when compressing earlier texts, namely T_1_ or T_2_. Therefore, the text T_3_ is compressed more effectively after text with the same source of information was compressed before it. The following simple example explains the essence of this method: Let T_1_ be a text in English, T_2_ in German, and (unknown) T_3_ also in English. Then d(T_1_T_3_) − d(T_1_) will be less than d(T_2_T_3_) − d(T_2_) because, in the first case, T_3_ in T_1_T_3_ was compressed after the archiver had been “tuned” to “its” statistics (for example, in the case of texts in English and German, the method works flawlessly with text lengths of several hundred letters for T_1_, T_2_ and T_3_).

This idea was proposed by Tehan [14,15] and was further developed by Ryabko and Savina (RS-method) [9,10,11]. In particular, in [9], this idea was applied to construct a statistical method for classifying texts, allowing one to determine the reliability of the obtained conclusions using mathematical statistics methods. The described scheme was also successfully applied by the authors of this paper to solve problems of text attribution in works [10], where it was experimentally shown that the individual style of an author can be determined quite accurately based on 4 KB of their text (approximately two pages of text in Russian or English). Based on this fact, we will apply the same scheme to solve the problem of recognizing the author’s style of writers of different language groups.

### 2.2. Description of the RS-Method for Recognizing the Author’s Style of a Writer

In order to make the description more understandable, we will illustrate it with an example of constructing a method for determining the author’s style of English-language writers. Let N writers and their works T_1_, T_2_, …, T_N_ be given.

Each text T_i_ is represented as two samples, called training (X_i_, i = 1, …, N) and experimental, which, in turn, consist of M parts (slices), which we will denote by Y_ij_, i = 1, …, N; j = 1, …, M.

For the experimental work, we compiled a sample of texts from Beresford, Jerome, Defoe and Locke, N = 4, M = 16. From the works of these authors, we made 4 training samples X_1_, …, X_4_, each 64 KB in size. Then we made test samples—16 files Y_1j_, j = 1, …, 16, each 4 KB in size, from the works of Beresford, Y_2j_, j = 1, …, 16, from the works of Defoe, and …, Y_16j_, j = 1, …, 16, from the works of Jerome and Locke. Then the file Y_1,1_ was successively “compressed” with the training samples of the sample X_1_, …,X_4_ and it was determined which of them was “better” compressed (i.e., d(X_1_ Y_1,1_) − d(X_1_), …, d(X_4_ Y_1,1_) − d(X_4_) were calculated and i was found, for which d(X_i_ Y_1,1_) − d(X_i_) is minimal). All Y_ij_, i = 1,…, 4; j = 1, …, 16, were processed similarly.

Table 1 presents the obtained data for the LZMA archiver, with a training sample (X_i_) of 64 kB and a slice (Y_ij_) of 4 kB.

Let us explain the meaning of these numbers: 16 in the upper left corner means that out of 16 files Y_1j_, j = 1, …, 16, all were compressed better with X_1_ (in other words, all 16 slices from Defoe’s works were compressed better with the training set of his works. The obtained result shows that D. Defoe’s author’s style is uniquely recognized by a 4 KB slice with a training set of 64 KB). The numbers from the first line mean that out of 16 files Y_2j_, j = 1, …, 16, 14 slices were compressed better with X_2_ (i.e., 14 slices from Beresford’s works were compressed better with his training set; however, 1 slice was more similar to Jerome’s works and 1 slice was similar to Locke’s works; here, the recognition of the writer’s style is 14 out of 16).

We will call the entire process of transition from the source texts T_1_, T_2_, …, T_N_ to the table (of size N × N) the construction of a contingency table, and we will denote the contingency table itself as W (T_1_, T_2_, …, T_N_) or W (depending on the context) and represent this table as follows:
        t_1,1_ t_1,2_ … t_1,N_W(T_1_, T_2_, …, T_N_) = t_2,1_ t_2,2_ … t_2,N_…………………        t_N,1_ t_N,2_ … t_N,N_

In addition, for each W table, we calculated the value of Cramer’s coefficient V [16]); here it should be noted that V is used to assess the relationship, or interdependence, and it takes values from zero to one, and a higher value indicates a greater dependence or interrelationship.

We will explain its meaning in more detail together with the contingency table W. As we saw in the example, the numbers in the cells of the contingency table indicate the number of slices whose authorship was attributed to a specific writer. If the method works “correctly”, i.e., it correctly determines the author’s style by the slices, then the values in the table will be concentrated mainly on the main diagonal. Otherwise, when the slices do not reveal the author’s style of the writer, the values in the table will be evenly distributed among different cells related to different writers.

This effect can be quantified using the Cramer V coefficient [14], which is calculated as follows: first, calculate P = $\sum_{i=1}^{N}\sum_{j=1}^{N}t_{ij},p_{ij}=\frac{t_{ij}}{P},p_{i}.=\sum_{j=1}^{N}p_{ij},{p.}_{j}=\sum_{i=1}^{N}p_{ij},$ and then calculate the following: $x^{2}$ = $\sum_{i=1}^{N}\sum_{i=1}^{N}\left({\left(t_{ij}-Np_{i}.{p.}_{j}\right)}^{2}/(Np_{i}.{p.}_{j})\right)$ and Cramer’s coefficient V = $\sqrt{x^{2}/(P(N-1))}$.

For Table 1, Cramer’s coefficient V = 0.9.

Note that the Cramer coefficient V = 1 if all nondiagonal elements are equal to 0, and V is equal to 0 if all t_i,j_ are equal.

Now let us pay attention to the choice of archiver. There are quite a lot of them at present. For this purpose, we examined the BZIP2, DEFLATE and LZMA archivers on the same sample. It turned out that the LZMA archiver has the highest Cramer coefficient; henceforth, we used this archiver. In our experiments, compression was performed using the 7-Zip archiver; the reference implementation of LZMA was developed in [17]. (We will not describe this in detail, since similar calculations were performed in [11], see 2.3. “Selection of method parameters”.)

## 3. Recognizing the Author’s Style of Writers in Different Language Groups

For our study, we selected 4 languages from different language groups belonging to different language families:

English (West Germanic language group of the Indo-European language family);

Amharic (Southern Ethiosemitic group of the Semitic language family);

Russian (East Slavic group of the Indo-European language family);

Chinese (Sinitic group of the Sino-Tibetan language family).

We note that we had already worked with texts in Russian and English in previous studies on determining the quality of translations and attribution of literary texts [10,11]. Therefore, we started with English. For this study, we selected the texts of the following works in English (see Table 2).

From each literary work, text pieces of 64 KB were taken for the training sample and text pieces of 64 KB for the test sample. Each sample was divided into 16 fragments (4 KB slices). Each fragment was added to the training sample in turn; the number of recognized fragments was recorded in the table. The results are presented in Table 3. The writers are presented by numbers.

The table shows that only two writers, Humphry Ward and Schreiner Olive, each had one slice attributed to the style of another writer. In George Eliot’s texts, two fragments out of 16 were attributed to Kipling. All other writers had their author styles recognized absolutely correctly: 16 slices out of 16. And the Cramer coefficient is close to 1 and equals V = 0.992.

To study the author style of Russian-speaking writers, 16 literary works by Russian writers of the late 19th–early 20th centuries were selected (see Table 4).

The preprocessing work with the Russian texts was exactly the same as with the English novels: a 64 KB sample, divided into 16 fragments of 4 slices. These 16 fragments of 4 KB were added to the training sample one by one for compression. The number of recognized slices was recorded in a table. The results are in Table 5.

The table shows a well-built diagonal consisting of recognized fragments of the author’s styles. However, Valery Bryusov’s texts were recognized in 10 fragments out of 16. This phenomenon has its own explanation. Bryusov is an outstanding Russian poet and the founder of Russian symbolism. His historical novel *The Altar of Victory* was the first prose work of the outstanding poet. The novel was dedicated to the Roman Empire during the era of its collapse. Apparently, his author’s style had not yet been formed; it contained many imitations and quotations. Bryusov used citations from 34 ancient poetic sources of various lyrical genres, and also accompanied the novel with notes occupying more than 100 of the 400 pages of text.

At the next stage of our research, we turned to the Chinese language. Chinese is a unique language with a rich history. It has features that are not found in other languages. As Chinese language experts note the Chinese language consists of many idioms [18,19]. An idiom is a stable figure of speech used as a single whole, forming a phraseological fusion [19]. An idiom can consist of 1–4 hieroglyphs. Each of the hieroglyphs carries its own semantic load, forming one image figure of speech [18]. An idiom is one indivisible lexical unit. Literary texts contain a large number of idioms. Idioms are written in hieroglyphs. Chinese writing is a logographic writing system in which symbols (logogram-hieroglyphs) [18,19] represent whole words or morphemes, but not individual sounds and letters [19]. Unlike phonetic writing, each hieroglyph is assigned not only a phoneme, but also a meaning, so the number of signs in Chinese writing is very large [20]. For our study, we selected literary works written in the official language, Putonghua (Mandarin) (see Table 6).

The texts were processed using the method already tested in English and Russian. We prepared a training sample of 64 KB and a test experimental sample of 64 KB. Then, for compression, 4 KB text fragments were added to the training sample, of which we selected 16. The results are presented in Table 7.

As can be seen from the table, the results are very similar to the results of the analyses of texts in Russian and English. The perfectly constructed diagonal shows the recognition of the author’s style of all writers.

The next language chosen for our study was Amharic. Amharic (አማርኛ) is the language of the Amhara people; it belongs to the Semitic family of languages [21]. For many years, Amharic was the official language of Ethiopia; now it has the status of being the working language of the government. About 25 million people in Ethiopia speak Amharic. The language is also widespread among some of the peoples of neighboring states: in Eritrea, Somalia, and Sudan [21]. It should be noted that more than 3 million emigrants speak Amharic outside of Ethiopia in the USA, Canada, Sweden and Israel. Amharic is used in business communications, in government agencies and in education. Newspapers, magazines and books are published in it. The list of literary works selected for the study is presented in Table 8.

Preliminary work with Amharic texts was the same as with other languages presented in the study. Two samples of 64 KB texts were formed: a training sample and a test experimental sample. Both samples were divided into 16 fragments of 4 KB. Then, 4 KB slices from the test sample were added one-by-one to the training sample for compression. After text compression, the results were entered into a table. The results are presented in Table 9.

The results of the study show that the RS-method of recognizing author’s style also works in the Amharic language. The Amharic language is a unique language that has a number of specific features. For example, the Amharic alphabet consists of 28 consonants and 7 vowels, but the writing system has special signs and combinations that bring the number of sounds to 200 [21]. The Amharic alphabet, also known as the Ethiopian script, is a syllabic script in which each sign represents a combination of a consonant and a vowel [22]. Despite the uniqueness and complexity of the Amharic language, Cramer’s coefficient is almost the same as that of the other languages we have considered.

## 4. Conclusions

The conducted study on the corpora of texts in different languages from four language groups showed that it is quite possible to determine author’s style using the RS-method. The main finding of the study (which was not known before) is the discovery of a new scientific fact: the same amount of data is required to recognize the author’s style of a writer in different languages that are culturally, historically and grammatically distant from each other. A completely natural question arises about the stability of these conclusions given different volumes of the training sample and sizes of the “slice”. It is natural to assume that with an increase in each of these parameters, and with their joint increase, the Cramer coefficient should increase. We deliberately conducted experiments on different sample sizes, similar to the one described above, and the results confirmed this assumption (see Table 10).

As can be seen from the table, the degree of change in the values of the Cramer coefficient remains approximately the same for all the languages considered, which confirms the conclusion that the amount of data required to recognize the author’s style in different languages from different language groups is almost the same or invariant.

Let us now discuss the possible applications of the developed method. Some of them are practical, while others are theoretical and even philosophical in nature.

Among the practical tasks, we will mention the detection of plagiarism and the determination of authorship. Among the theoretical tasks are issues related to artificial intelligence systems being capable of maintaining dialogue with people and/or creating texts on specific topics. An interesting question is whether different artificial intelligence systems have their own authorial style. And if so, is it possible to build an artificial intelligence system without an authorial style (or with a hidden authorial style)? Another related question is whether there is a certain level of complexity needed for a system to be capable of maintaining a dialogue with a human being, above which the system must have its own unique style. Perhaps the approach proposed here could become a tool for investigating such problems.

## Figures and Tables

**Table 1 entropy-27-01039-t001:** Recognition of the author’s style of the writers.

Writers	Beresford	Defoe	Jerome	Locke
John Davys Beresford	14	0	1	1
Daniel Defoe	0	16	0	0
Jerome Klapka Jerome	1	0	14	1
John Locke	0	0	1	15

**Table 2 entropy-27-01039-t002:** List of literary works in English selected for the study.

No.	Author Names	Book Titles	Published Year
1.	Florence L. Barclay	*The White Ladies of Worcester*	1917
2.	Arnold Bennet	*Imperial Palace*	1930
3.	R. D. Blackmore	*A Tale of the Great War*	1887
4.	Frances Hodgson Burnett	*The Secret Garden*	1911
5.	Gilbert Keith Chesterton	*The Innocence of Father Brown*	1911
6.	Arthur Conan Doyle	*The Lost World*	1912
7.	George Eliot	*Felix Holt, the Radical*	1866
8.	Ford Madox Ford	*The Good Soldier*	1915
9.	John Galsworthy	*Over the River*	1933
10.	George Gissing	*Will Warburton*	1903
11.	Rudyard Kipling	*Kim*	1901
12.	D. H. Lawrence	*Women in Love*	1920
13.	Humphry Ward	*Harvest*	1920
14.	Virginia Woolf	*To the Lighthouse*	1927
15.	Schreiner Olive	*Undine*	1929

**Table 3 entropy-27-01039-t003:** Recognition of authorial styles of English writers.

V = 0.992															
	1	2	3	4	5	6	7	8	9	10	11	12	13	14	15
1	16	0	0	0	0	0	0	0	0	0	0	0	0	0	0
2	0	16	0	0	0	0	0	0	0	0	0	0	0	0	0
3	0	0	16	0	0	0	0	0	0	0	0	0	0	0	0
4	0	0	0	16	0	0	0	0	0	0	0	0	0	0	0
5	0	0	0	0	16	0	0	0	0	0	0	0	0	0	0
6	0	0	0	0	0	16	0	0	0	0	0	0	0	0	0
7	0	0	0	0	0	0	14	0	0	2	0	0	0	0	0
8	0	0	0	0	0	0	0	16	0	0	0	0	0	0	0
9	0	0	0	0	0	0	0	0	16	0	0	0	0	0	0
10	0	0	0	0	0	0	0	0	0	16	0	0	0	0	0
11	0	0	0	0	0	0	0	0	0	0	16	0	0	0	0
12	0	0	0	0	0	0	0	0	0	0	0	16	0	0	0
13	0	0	0	0	0	1	0	0	0	0	0	0	15	0	0
14	0	0	0	0	0	0	0	0	0	0	0	0	0	16	0
15	0	0	0	0	0	0	1	0	0	0	0	0	0	0	15

**Table 4 entropy-27-01039-t004:** List of literary works by Russian writers selected for research.

No.	Author Names	Book Titles	Published Year
1.	Mikhail Bulgakov	*The White Guard*	1925
2.	Anton Chekhov	*Lady with a Dog*	1898
3.	Fyodor Dostoevsky	*Demons*	1872
4.	Fyodor Sologub	*Drops of Blood*	1905
5.	Valery Bryusov	*Altar of Victory*	1912
6.	Zinaida Gippius	*Devil’s Doll*	1911
7.	Nikolai Gogol	*Dead Souls*	1842
8.	Maxim Gorky	*The Artamonov Business*	1925
9.	Alexander Herzen	*Who is to Blame?*	1846
10.	Vladimir Nabokov	*The Gift*	1938
11.	Avdotya Panaeva	*The Talnikov Family*	1928
12.	Alexander Pushkin	*The Captain’s Daughter*	1836
13.	Lev Tolstoy	*Resurrection*	1899
14.	Ivan Turgenev	*On the Eve*	1860
15.	Mikhail Lermontov	*Hero of our time*	1840
16.	Maria Zhukova	*Dacha on Peterhof Road*	1845

**Table 5 entropy-27-01039-t005:** Recognition of authorial styles of Russian writers.

V = 0.993																
	1	2	3	4	5	6	7	8	9	10	11	12	13	14	15	16
1	16	0	0	0	0	0	0	0	0	0	0	0	0	0	0	0
2	0	14	0	0	0	1	0	0	0	0	0	0	0	1	0	0
3	0	0	16	0	0	0	0	0	0	0	0	0	0	0	0	0
4	0	0	0	16	0	0	0	0	0	0	0	0	0	0	0	0
5	0	0	0	0	10	0	0	0	0	0	0	1	0	0	1	4
6	0	0	0	0	0	16	0	0	0	0	0	0	0	0	0	0
7	0	0	0	0	0	0	16	0	0	0	0	0	0	0	0	0
8	0	0	0	0	0	0	0	16	0	0	0	0	0	0	0	0
9	0	0	0	0	0	0	0	0	16	0	0	0	0	0	0	0
10	0	0	0	0	0	0	0	0	0	16	0	0	0	0	0	0
11	0	0	0	0	0	0	0	0	0	0	16	0	0	0	0	0
12	0	0	0	0	0	0	0	0	0	0	0	16	0	0	0	0
13	0	0	0	0	0	0	0	0	0	0	0	0	15	1	0	0
14	0	0	0	0	0	0	0	0	0	0	0	0	0	16	0	0
15	0	0	0	0	0	0	0	0	0	0	0	0	0	0	16	0
16	0	0	0	0	0	0	0	0	0	0	0	0	0	0	0	16

**Table 6 entropy-27-01039-t006:** List of authors and works in Chinese.

No.	English Author Name	English Title	Chinese Author Name	Chinese Title	First Published	Revised Edition
1	Lao She	*Rickshaw Boy*	老舍	骆驼祥子	1939	2010
2	Liu Cixin	*The Three-Body Problem*	刘慈欣	三体	2008	2020
3	Zhang Ailing (Eileen Chang)	*Half a Lifelong Romance*	张爱玲	半生缘	1951	2014
4	Lu Xun	*Call to Arms*	鲁迅	呐喊	1923	2000
5	Qian Zhongshu	*Fortress Besieged*	钱钟书	围城	1947	2003
6	Lu Yao	*Ordinary World*	路遥	平凡的世界	1986	2017
7	Yu Hua	*To Live*	余华	活着	1993	2014
8	Wang Anyi	*The Song of Everlasting Sorrow*	王安忆	长恨歌	1996	2008
9	Mo Yan	*Life and Death Are Wearing Me Out*	莫言	生死疲劳	2006	2012
10	Jia Pingwa	*The Qin Opera*	贾平凹	秦腔	2005	2016
11	Jin Yucheng	*Blossoms*	金宇澄	繁花	2013	2023
12	Shen Congwen	*Border Town*	沈从文	边城	1934	2009
13	A Lai	*Red Poppies: A Novel of Tibet*	阿来	尘埃落定	1998	2002
14	Chen Zhongshi	*White Deer Plain*	陈忠实	白鹿原	1993	2023
15	Chi Zijian	*The Last Quarter of the Moon*	迟子建	额尔古纳河右岸	2005	2013

**Table 7 entropy-27-01039-t007:** Recognition of the author’s styles of Chinese writers.

Cramer’s V = 0.984															
	1	2	3	4	5	6	7	8	9	10	11	12	13	14	15
1	16	0	0	0	0	0	0	0	0	0	0	0	0	0	0
2	0	16	0	0	0	0	0	0	0	0	0	0	0	0	0
3	0	0	16	0	0	0	0	0	0	0	0	0	0	0	0
4	0	0	4	12	0	0	0	0	0	0	0	0	0	0	0
5	0	0	0	0	16	0	0	0	0	0	0	0	0	0	0
6	0	0	0	0	0	16	0	0	0	0	0	0	0	0	0
7	0	0	0	0	0	0	16	0	0	0	0	0	0	0	0
8	0	0	0	0	0	0	0	16	0	0	0	0	0	0	0
9	0	0	0	0	0	0	0	0	16	0	0	0	0	0	0
10	0	0	0	0	0	0	0	0	0	16	0	0	0	0	0
11	0	0	2	0	0	0	0	0	0	0	14	0	0	0	0
12	0	0	0	0	0	0	0	0	0	0	0	16	0	0	0
13	0	0	0	0	0	0	0	0	0	0	0	0	16	0	0
14	0	0	0	0	0	0	0	0	0	0	0	0	0	16	0
15	0	0	0	0	0	0	0	0	0	0	0	0	0	0	16

**Table 8 entropy-27-01039-t008:** List of authors and literary works in Amharic.

No.	Author Name in English	Book Titles in English	Author Name in Amharic	Book Titles in Amharic	Published Year
1.	Haddis Alemayehu	*Love to the Grave*	ሀዲስ አለምየሁ	ፍቅር እስከ መቃብር	1968
2.	Bealu Girma	*Oromaye*	በዓሉ ግርማ	ኦሮማይ	1983
3.	Mammo Wudneh	*Eye of the Needle*	ማሞ ውድነህ	ሾተላይ	1981
4.	Tsegaye Gabre-Medhin	*Makbeth*	ጸጋዬ ገብረመድህን	ማክቤዝ	1972
5.	Kebede Michael	*A prophetic Appointment*	ከበደ ሚካኤል	የትንቢት ቀጠሮ	1959
6.	Alemayehu Wassie	*Emegoa*	አለማየሁ ዋሴ	እመጎ	2008
7.	Sahle Sellassie Berhane Mariam	*Mr. Ketaw*	ሣህለሥላሴ ብርሃነማርያም	ባሻ ቅጣው	1976
8.	Adam Reta	*Mehalet*	አዳም ረታ	ማህሌት	2002
9.	Tekletsadik Mekuria	*Emperor Menilik and Ethiopian Unity*	ተክለ ጻድቅ መኩሪያ	ዐፄ ምኒልክ እና የኢትዮጵያ አንድነት	1983
10.	Muluken Tariku	*Emperor Minilik and Adwa victory*	ሙሉቀን ታሪኩ	አፄ ምኒልክ እና የአድዋ ድል	2006
11.	Afework Gebereyesus	*Tobia*	አፈወርቅ ገ/ኢየሱስ	ጦቢያ	1900
12.	Aleqa Taye	*Ethiopian History*	አለቃ ታየ	የኢትዮጵያ ህዝብ ታሪክ	1914
13.	Bahru Zewde	*Modern Ethiopia*	ባህሩ ዘውዴ	ዘመናዊ የኢትዮጵያ ታሪክ	1999
14.	Berhanu Zereyehun	*The Tear of Tewodros*	ብርሃኑ ዘርይሁን	የቴድሮስ ዕምባ	1960

**Table 9 entropy-27-01039-t009:** Results of recognition of the author’s styles of writers in Amharic.

Cramer’s V = 0.892														
	1	2	3	4	5	6	7	8	9	10	11	12	13	14
1	16	0	0	0	0	0	0	0	0	0	0	0	0	0
2	0	14	2	0	0	0	0	0	0	0	0	0	0	0
3	0	0	16	0	0	0	0	0	0	0	0	0	0	0
4	0	0	0	15	1	0	0	0	0	0	0	0	0	0
5	0	0	0	0	16	0	0	0	0	0	0	0	0	0
6	0	2	0	0	1	13	0	0	0	0	0	0	0	0
7	0	0	0	0	0	0	16	0	0	0	0	0	0	0
8	0	1	1	0	0	1	0	13	0	0	0	0	0	0
9	0	0	0	0	0	0	0	0	16	0	0	0	0	0
10	0	0	0	0	0	0	0	0	6	9	0	0	1	0
11	0	0	0	0	0	0	0	0	0	0	16	0	0	0
12	0	2	0	0	0	0	0	0	0	0	0	14	0	0
13	0	0	0	0	0	0	0	0	0	0	0	0	16	0
14	0	0	0	0	0	0	0	0	0	0	0	0	0	16

**Table 10 entropy-27-01039-t010:** Parameters’ efficiency comparisons.

Parameters	Language	Training Sample Size	Test Sample Size	Cramer V
Parameter 1	Amharic	96	8	0.928
	English			1
	Russian			1
	Chinese			1
	Amharic	96	4	0.914
	English			0.994
	Russian			0.991
	Chinese			0.997
	Amharic	96	2	0.916
	English			0.983
	Russian			0.976
	Chinese			0.978
Parameter 2	Amharic	64	8	0.919
	English			1
	Russian			1
	Chinese			0.992
	Amharic	64	4	0.892
	English			0.992
	Russian			0.993
	Chinese			0.984
	Amharic	64	2	0.873
	English			0.98
	Russian			0.97
	Chinese			0.979
Parameter 3	Amharic	48	8	0.913
	English			1
	Russian			1
	Chinese			0.971
	Amharic	48	4	0.912
	English			0.990
	Russian			0.981
	Chinese			0.960
	Amharic	48	2	0.887
	English			0.942
	Russian			0.982
	Chinese			0.952

## Data Availability

The original contributions presented in this study are included in the article. Further inquiries can be directed to the corresponding author.

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
