# Peer review of "The Amount of Data Required to Recognize a Writer’s Style Is Consistent Across Different Languages of the World"

_entropy, 2025, doi:10.3390/e27101039_

Round 1
Reviewer 1 Report
Comments and Suggestions for Authors
The main contribution of the paper is the conclusion that the amount of data required to recognize the style of writing of an author by using the method proposed by the first two authors of this paper does not depend on the language in which the text processed is written. The experiment is performed over texts in four different language groups and families. The amount of data is measured in kilobytes, and not in the number of letters or symbols.
The authors determined the Cramer coefficient for different sizes of the training and test samples. The degree of changes of this coefficient supports the conclusion about invariance of the amount of data.
Could any comment be given on what is the reason for the above main conclusion in the paper? Is it related to the main feature of the RS-method which exploits the data compression and search for identical patterns?
Author Response
The authors would like to express sincere gratitude to the reviewer for his very important and useful comments. All comments have been taken into account.
Yes, the conclusion is based on properties of the RS-method.
Reviewer 2 Report
Comments and Suggestions for Authors
Section 2.1 - How proposed method differs from Normalized compression distance? This distance is missing in the state of the art. Please explain it. Is there any link to Kolmogorov complexity?
Section 2.2 - Why N = 4 and M = 16. Is there any reason for this values? Why LZMA method is used? What archiver did you used for this experiment? 7zip? There is no reference.
Table on lines 132 - 134 is example of very bad typography. Also typography of Cramer V at lines 147 - 150 has to be improved.
line 155 Why did the authors exclude the BZIP2 and DEFLATE methods from further investigation? There is no discussion.
Chapter 3 is full of annoying tables. Tables 2, 4, 6 and 8 presents list of texts used in experiments. Tables 3, 5, 7 and 9 presents experimental result, but there are only some numbers on diagonal and everything else is zero. Yes, these table presents experimental results, but they are nearly same. All these tables can be put in some kind of appendix or suplementary material. In the chapter 3 they just fill the space.
Unfortunately, very little space is devoted in this chapter to discussing the results, presenting results with other compression methods. And there is a complete lack of comparison with any existing methods solving a similar problem.
The paper has to be significantly improved and rewritten to be acceptable.
Reviewer 3 Report
Comments and Suggestions for Authors
From a Digital Humanities perspective, this is an interesting study. The application of the Kolmogorov complexity and compression distance to text clustering for literary studies is quite old now (at least 20 years), but the point that the authors want to make, namely that the amount of data required is similar across “different languages, culturally, historically, grammatically distant from each other” is innovative.
From a (computational) linguistic perspective, this raises questions. The languages themselves seem to differ in the compactness of the information (e.g. in an inflectional language such as Amharic one word contains more information than a word in English or Chinese). Does this effect the compression? The sizes of the corpora are expressed in KB. Are the underlying texts also of the same size? Or do the number of words of the size of the text differ (and how should they be measured, given the various tokens of the script in, e.g., Amharic (abiguda), English (alphabetic) and Chinese (partly pictographic?) (I assume the alphabetic system of Russian functions more or less like English). Is there room to address these questions, even if shortly?
Author Response
The authors would like to express sincere gratitude to the reviewer for his very important and useful comments. All comments have been taken into account.
We estimated the text size in KB, as this makes it possible to ignore the alphabet structure.
Round 2
Reviewer 2 Report
Comments and Suggestions for Authors
I am satisfied with your updates and answers. Good luck!